# Diverse Inference for Solving ARC at a Human Level

**AI assistants, Seunghwan Hyun** [1]**, Gaston Longhitano** [1]**, Mao Mao** [1]

**Yuke Zhang** [1]**, Ben Segev** [2]**, Iddo Drori** [3,4,5]

[1] Boston University, [2] Independent, [3] Yeshiva University, [4] Tel Aviv University, [5] Stanford University

## Abstract

Reasoning LLMs have made significant progress in mathematics and coding, yet struggle with advanced generalization tasks such as Abstraction and Reasoning Corpus (ARC) puzzles. To address this, we propose a diverse inference approach that aggregates multiple models and methods at test time to enhance the performance. We automatically verify correctness of solutions to ARC puzzles by code. Our approach increases the success rate on a validation set of 400 ARC puzzles from 53% to 69.5% without reasoning models and 91.5% to 93.75% with them which exceeds average human accuracy which is between 73.3 and 77.2%. Our approach succeeds in solving ARC puzzles that state-of-the-art reasoning LLMs and 948 humans could not. It solves 26.5% of ARC puzzles that reasoning models do not solve and 80% of ARC puzzles that 948 humans could not. We identify the relationship between the number of diverse models and methods and the performance on verifiable problems. We automate the MARC method by an agentic framework using a computation graph, enabling modular, scalable, and autonomous execution and comparison of ARC problem-solving pipelines. This work makes progress toward building flexible, generalizable reasoning systems.

## 1 Introduction

Reasoning Large Language Models (LLMs) have led to impressive performance in mathematics, coding, and problem-solving. Despite this progress, a single large model or method may struggle with challenging tasks. To address this, diversity of models and methods for inference has emerged as a mechanism to increase performance by using complementary strengths.

We demonstrate the advantages of diverse inference on the Abstraction and Reasoning Corpus (ARC) [1], which is a representative and challenging visual reasoning benchmark: We solve 80% of puzzles that 948 humans collectively could not solve.

Our key methodological contributions that drive these results are:

1. Diverse inference. We aggregate multiple models, methods, and agents at test time rather than relying on a single model or method. Any single correct solution is validated automatically for the verifiable tasks of ARC puzzles. For ARC tasks, synthesized code solutions are verified on training examples as unit tests.

2. Test-time simulations and reinforcement learning. We generate additional problem-specific information at inference time. We explore puzzle transformations by synthesizing code that prunes incorrect solutions and refines candidate solutions. Searching using trained verifiers often outperforms supervised fine-tuning given the same dataset [2], which motivates reinforcement learning fine-tuning. We run simulations and reinforcement learning at test time to synthesize additional data that allows us to solve difficult ARC puzzles.

1st Open Conference on AI Agents for Science (agents4science 2025).

## 1.1 Related Work

**Abstraction and Reasoning Corpus (ARC).** A benchmark introduced [1] to measure the visual reasoning aspect of artificial general intelligence by a set of puzzles with patterns on visual grids. Given a small set of training pairs of input and output, the goal is to infer the transformation, relationship, or function between them. To verify if the predicted transformation is accurate, the transformation logic is applied to a test example, and if the outputs match, the task is considered to be solved. The average human performance on ARC is between 73.3% and 77.2%, and it takes 948 humans to collectively solve 98.8% of the evaluation set puzzles correctly [3].

ARC consists of 400 public training tasks, 400 public evaluation tasks, and 200 private evaluation tasks, where the difficulty of public training tasks is 'easy' and other tasks are 'hard'. Because each task has different transformations and private evaluation tasks, which are not released to the public, and measures the performance of different models and methods, it should not be possible to prepare for any of the tasks. Each task has 2 or more training input-output pairs, with a median of 3. All the inputs and outputs are rectangular grids of variable size, which go up to 30 by 30. Each cell of a grid can be one of 10 different values or colors.

**From mixture of experts to diverse models and methods.** Most recent language models use a mixture of experts [4], where multiple experts are trained to specialize in different aspects of the input space. A gating mechanism learns to select or weigh the experts based on input. The diversity in expertise allows the model to use a broad range of problem-solving strategies, and distribution among diverse experts allows the model to handle variations better. Large-scale transformers that leverage diversity [5, 6] increase efficiency and accuracy, otherwise difficult to achieve with a single monolithic model. In this work, we use diverse models and methods to increase accuracy.

**Perfect, near-perfect, and imperfect verifiers.** An imperfect verifier fails to filter out false positives, which are wrong solutions that pass the verifier. These false positives impose an upper bound on accuracy despite the increase in sampling or inference time [7]. In this work, we use near-perfect verifiers. We use code execution on the training examples as near-perfect verifiers and compare the predicted output with the true output. This may be near perfect rather than perfect since there may be more than a single code solution for an ARC puzzle.

**Empirical scaling laws.** The two most common empirical scaling laws for foundation model performance are:

1. The relationship between model size, data size, and loss, i.e., language models with more parameters, training data, and training time perform better [8], quantified by OpenAI's scaling law [9] and the Chinchilla scaling law [10]. Scaling laws extend to fine-tuning, describing the relationship between model performance and the number of fine-tuning parameters and fine-tuning data size [11], and extend to different architectures and downstream tasks [12].

2. The relationship between model performance and test-time compute. The tradeoff between training time and test time compute has been demonstrated early on for board games [13], showing that increasing either one leads to better performance. Test time compute scaling [14] has been demonstrated again by DeepMind on coding [15] and OpenAI o1 [16] and o3-mini [17] for reasoning LLMs.

We identify a third empirical scaling law: The relationship between the number of diverse models and methods and the performance on verifiable problems.

**AI agents and skills.** Beyond building and evaluating AI agent [18, 19, 20] workflows, agents may be defined by their skills [21] to improve performance.

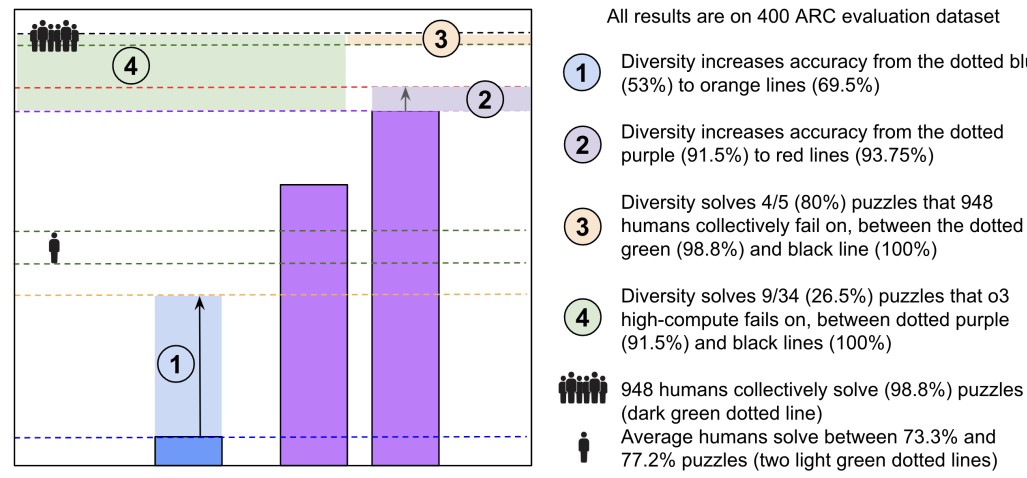

Figure 1: Zooming in on diversity performance of 16 models and methods on the 400 ARC evaluation puzzles.

## 2 Methods

### 2.1 Reasoning LLMs

A foundation model $\pi$ with pre-trained parameters $\theta$ defines a conditional distribution:

$$p_\theta(y \mid x), \tag{1}$$

where $x$ is a prompt and $y$ is a response. A reasoning model is trained to generate a (hidden) rationale, also known as chain-of-thought (CoT) $z$, so that the joint generation is given by:

$$p_\theta(z, y \mid x) = p_\theta(z \mid x)\, p_\theta(y \mid z, x). \tag{2}$$

Model training consists of two phases: (i) Supervised fine-tuning (SFT): from $\pi$ to $\pi_{\text{SFT}}$; and (ii) Reinforcement learning (RL): from $\pi_{\text{SFT}}$ to $\pi_{\text{RL}}$.

#### 2.1.1 Supervised fine-tuning (SFT)

Samples are generated using $\pi_\theta$ in Eq. 1 and stored in a dataset $\mathcal{D} = \{(x^i, y^i)\}_{i=1,\ldots,n}$. A supervised fine-tuning loss is derived by taking the negative log likelihood of Eq. 1 on the dataset:

$$\mathcal{L}(\theta) = - \sum_{(x^i, y^i)\, \in\, \mathcal{D}} \log p_\theta\big(y^i \mid x^i\big). \tag{3}$$

Similarly, for a reasoning model, samples are generated using $\pi_\theta$ in Eq. 2 and stored in a dataset $\mathcal{D} = \{(x^i, z^i, y^i)\}_{i=1,\ldots,n}$. A supervised fine-tuning loss is derived by taking the negative log likelihood of Eq. 2 on the dataset:

$$\mathcal{L}(\theta) = - \sum_{(x^i, z^i, y^i)\, \in\, \mathcal{D}} \Big[ \log p_\theta\big(z^i \mid x^i\big) + \log p_\theta\big(y^i \mid x^i, z^i\big) \Big]. \tag{4}$$

#### 2.1.2 Reinforcement learning

For tasks such as solving math problems or generating code, we define a reward function $R(x, y)$ that is checked automatically, by verifying an answer or proof, or by running unit tests. We then optimize:

$$\underset{\theta}{\text{maximum}}\; \mathbb{E}_{x \sim \mathcal{D},\, y \sim \pi_\theta} \big[ R(x, y) \big].$$

This is a classical RL objective without the need for a learned preference model.

More generally, given a foundation model, we define a reward:

$$r(x, \hat{y}) = f\big(\pi_{\text{RM}}(x, \hat{y})\big), \tag{5}$$

where $\hat{y}$ is the resulting output, and $f$ is a function measuring the quality of that output result. For example, using policy gradient, we update $\theta$ by:

$$\nabla_\theta \, \mathcal{L}_{\text{RL}} = - \, \mathbb{E}_{\hat{y} \sim \pi_\theta(\cdot|x)} \Big[ r\big(x, \hat{y}\big) \, \nabla_\theta \, \log \pi_\theta\big(\hat{y} \mid x\big) \Big]. \tag{6}$$

For a reasoning model, let $\hat{z}$ be a sampled rationale and define a reward [22]:

$$r(x, \hat{z}, \hat{y}) \;=\; f\big(\pi_{\text{RM}}(x, \hat{z}, \hat{y})\big), \tag{7}$$

where $f$ is a function quantifying the quality of the rationale, for example, the log-likelihood improvement on future tokens as a reward, or correctness on a question answering task. For a reasoning model, plugging in the logarithm of Eq. 2:

$$\log p_\theta(\hat{z}, \hat{y} \,|\, x) = \log p_\theta(\hat{z} \,|\, x) + \log p_\theta(\hat{y} \mid x, \hat{z}), \tag{8}$$

yields the gradient:

$$\nabla_\theta \, \mathcal{L}_{\text{RL}} = - \, \mathbb{E}_{\hat{z}, \hat{y} \sim \pi_\theta(\cdot|x)} \Big[ r\big(x, \hat{z}, \hat{y}\big) \, \nabla_\theta \, \log \pi_\theta(\hat{z} \,|\, x) \\ + \log \pi_\theta(\hat{y} \,|\, x, \hat{z}) \Big]. \tag{9}$$

## 2.2 Diverse Models and Methods

We ablate multiple models and methods at test time using inference methods implemented by OptiLLM [23], BARC [24], and MARC [25]:

- **Zero-shot**: Zero-shot approach in LLM research represents the basic methodology. The problem $x$ is given to the LLM $f$ as-is without additional context information or training data. Output is simply the answer of LLM, denoted as $f(x)$.

- **Best of $N$ sampling**: This simple method is often used in generative models to select the best answer among multiple candidates. Given $n$ candidate responses $Y = \{y^1, y^2, \ldots, y^n\}$ this method selects the best one based on a criterion $y^* = \arg\max_{y^j \in Y} C(y^i)$ where $C(y^i)$ is a scoring function. Given a verifier and a chain of thought, we perform rejection sampling, by sampling different chains of thought $z^i \sim p(z \mid x)$, their responses $y^i \sim p(y \mid x, z^i)$ and keeping those responses $y^i$ that are verified.

- **MCTS** [26]: Monte Carlo Tree Search (MCTS) is a search algorithm that explores the search space. It gained popularity with success in games with very large search spaces, such as Chess and Go, by proving its ability to effectively balance exploration and exploitation. Algorithm will select the best child node based on the node value which is estimated by $V(s) = \frac{1}{N(s)} \sum_{i=1}^{N(s)} R_i$, where $N(s)$ is the number of times node $s$ has been visited and $R_i$ is the reward from simulation $i$. Then it generates new child nodes to expand a search tree and a new node, the algorithm randomly chooses actions until reaching a terminal state, and obtains a reward $R_i$. In this study, we perform rejection sampling from an intermediate step in the chain of thought by Monte-Carlo simulations.

- **Self-consistency** [27]: This technique boosts the performance of Chain-of-Thought reasoning in large language models (LLMs). Instead of relying on a single response, self-consistency evaluates multiple outputs $Y = \{y^1, y^2, \ldots, y^n\}$ for the same input $x$ and selects the most common or majority vote response $y^* = \text{Majority Vote}(\{Y\})$ at intermediate steps. This approach enhances the reliability and accuracy of predictions, reducing variability and improving the overall quality of the model's output; however, it often saturates with sufficient samples.

- **Mixture of agents** [28]: Mixture of agents (MoA) leverages collective strengths of multiple agents or LLMs. This can be further applied to integrating different agents specifically trained or designed for given tasks. The paper uses an example with multiple layers where each layer contains multiple agents or models, $M = \{m^1, m^2, \ldots, m^k\}$. In each layer $j$, MoA stores generate outputs $p^i = m^i(x^j)$ for an input $x^j$. $\{p^1, p^2, \ldots, p^n\}$ are aggregated using the Aggregate-and-Synthesize prompt which outputs $a^j$. $a^j$ is concatenated to input prompt $x^j$ and becomes $y^j$. $y^j$ becomes $x^{j+1}$ and is processed as input for the next layer.

- **Plan search (PS)** [29]: This search method enhances LLM's performance by generating a diverse set of observations about a problem and using them to create plans through combination. Searching through different plans in natural language instead of code solutions, Plan search is able to explore a significantly broader idea space. Then, using each plan, candidate codes are generated and then evaluated to select the best solution.

- **BARC** [24]: This framework combines induction and transduction methods to solve ARC puzzles. Each puzzle is comprised of pairs of input $x$ and output $y$ mapped from the latent function $y_{train} = f(x_{train})$. Induction infers the latent function $f$ where transduction directly predicts the $y_{test}$ from given $x_{train}, y_{train}, x_{test}$. To combine the output of induction and transduction models, we check if the inferred $f_{inferred}$ is valid by checking if $f_{inferred}(x_{train})$ matches $y_{train}$. If the solution is plausible, that becomes the predicted $y_{test}$. Otherwise, predicted $y_{test}$ is the output of the transduction model.

- **MARC** [25]: Using test-time training, which increases performance by generating a dataset by leave-one-out and rule-based augmentations. This data augmentation allows models to leverage in-context learning for each puzzle given a sequence of input-output pairs $\{x_1, y_1, \ldots x_n, y_n, x_{n+1}\}$ where the model generates the predicted output $\hat{y}_{n+1}$ by sampling from $\hat{y}_{n+1} \sim p(\cdot | x_1, y_1, \ldots x_n, y_n, x_{n+1})$.

## 2.3 Aggregating Diverse Models and Methods

We aggregate the results of diverse models and methods whose solutions may be perfectly verified as correct by a maximum. Let $\mathcal{T} = \{t_1, t_2, \ldots, t_N\}$ be the set of $N$ ARC problems and $K$ the number of models and methods $\mathcal{M} = \{\mathcal{M}_1, \mathcal{M}_2, \ldots, \mathcal{M}_K\}$, where each $\mathcal{M}_k \in \mathcal{M}$ attempts to solve each $t_i \in \mathcal{T}$. The indicator is defined by:

$$\mathbb{1}\big(\mathcal{M}_k \text{ solves } t_i\big) = \begin{cases} 1, & \text{if } \mathcal{M}_k \text{ correctly solves } t_i, \\ 0, & \text{otherwise.} \end{cases}$$

Since we can verify the correctness of each individual solution, for each problem $t_i$, there exists a ground truth validation mechanism indicating whether $\mathcal{M}_k$'s proposed solution is correct. We combine the outputs of all models by taking the logical maximum, i.e., logical OR, over their correctness indicators: $\mathbb{1}\big(\text{any model solves } t_i\big) = \max_{k \in \{1, \ldots, K\}} \mathbb{1}\big(\mathcal{M}_k \text{ solves } t_i\big)$. Problem $t_i$ is considered solved if and only if at least one method in $\mathcal{M}$ succeeds in solving it. We define the success rate, or accuracy, of the aggregated system across the set $\mathcal{T}$ of $N$ problems as: $\frac{1}{N} \sum_{i=1}^{N} \max_{k \in \{1, \ldots, K\}} \mathbb{1}\big(\mathcal{M}_k \text{ solves } t_i\big)$. Since a problem is counted as solved if any one of the $K$ models or methods solves it, this aggregation is the best-case scenario. If all models make different systematic approaches, it will substantially improve the coverage of solvable problems relative to individual models. If any model's solution is correct for a particular problem, that problem is marked as solved in the aggregated result, giving the maximum performance across diverse models.

# 3 Results

## 3.1 Summary

We perform an extensive evaluation of 16 models and methods on 400 ARC evaluation puzzles as illustrated in Figures 2, 1, and Table 3. Diversity is the maximum verifiable aggregation of 16 models and methods at inference time. We find that:

1. Without reasoning LLMs, diversity of 16 models and methods increases performance from the blue dotted line (53%) to the orange dotted line (69.5%).

2. With reasoning LLMs, diversity of 16 models and methods increases performance from the purple dotted line (91.5%) to the red dotted line (93.75%).

3. Diversity of 16 models and methods solves 26.5% of the puzzles on which reasoning LLMs fail. These 34/400 puzzles are between the dotted purple line (91.5%) and the black line (100%).

4. Diversity of 16 models and methods solves 80% of the puzzles on which 948 humans collectively fail. These 5/400 puzzles are between the dotted green line (98.8%) and the black line (100%).

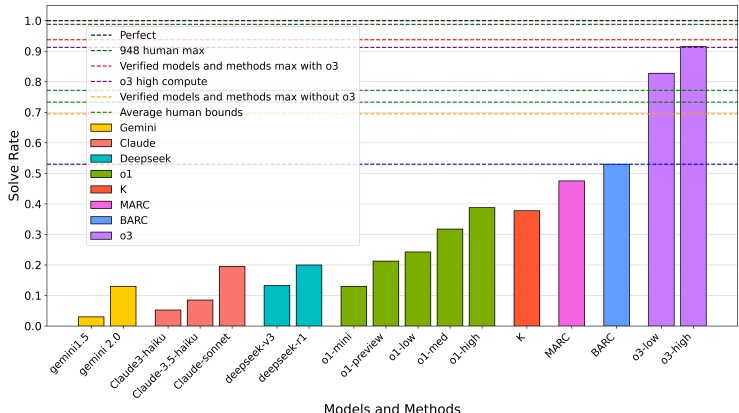

Figure 2: ARC performance for different models and methods and human performance on the evaluation dataset of 400 puzzles.

## 3.2 Diverse Model and Method Success on Failure Cases of o3-high

Figures 3, 4, 5, and 6 show results of tasks that o3-high failed to solve using different methods and models. For each method and model, Table 1 reports if the answer is correct by ✓, and ×otherwise. Running times, in brackets, are in seconds. Average running times are between 99 and 593 seconds.

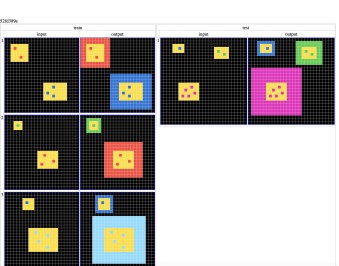

Figure 3: ARC task 52fd389e on which o3 high compute fails and another model or method succeeds.

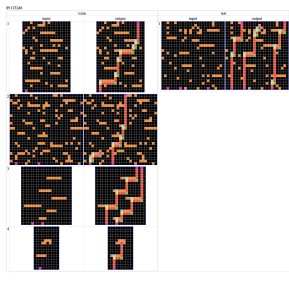

Figure 4: ARC task 891232d6 on which o3 high compute fails and another model or method succeeds.

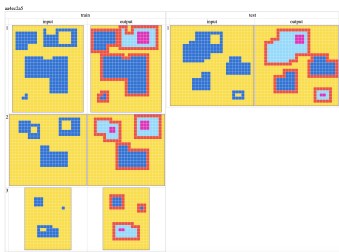

Figure 5: ARC task aa4ec2a5 on which o3 high compute fails and another model or method succeeds.

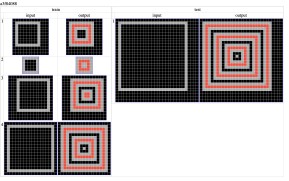

Figure 6: ARC task a3f84088 on which o3 high compute fails and another model or method succeeds.

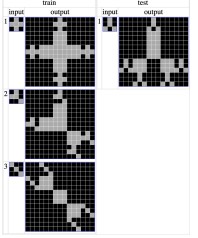

Figure 7: ARC task 8719f442 on which 948 humans fail and a model or method succeeds.

## 3.3 Diverse Model and Method Success on Failure Cases of 948 Humans

Figures 10, 7, 8, and 9 show results of tasks that 948 humans failed to solve using different methods and models. For each method and model, Table 2 reports if the answer is correct by ✓, and ×otherwise.

Table 1: Ablation experiments on difficult ARC problems on which o3 high compute fails on.

| ARC o3h × | max | cs | o1h | v3 | r1 | MCTS | BoN | MoA | SC | PS | BARC | MARC |
|---|---|---|---|---|---|---|---|---|---|---|---|---|
| 05a7bcf2 | × | × | × | × | × | ×(152) | ×(113) | ×(451) | ×(561) | ×(79) | ×(268) | ×(580) |
| 0934a4d8 | × | × | × | × | × | ×(188) | ×(160) | ×(328) | ×(382) | ×(86) | ×(76) | ×(240) |
| 09c534e7 | × | × | × | × | × | ×(177) | ×(178) | ×(458) | ×(453) | ×(182) | ×(193) | ×(271) |
| 0d87d2a6 | ✓ | × | × | × | × | ×(181) | ×(90) | ×(410) | ×(425) | ×(102) | ✓(110) | ×(246) |
| 1acc24af | × | × | × | × | × | ×(125) | ×(67) | ×(236) | ×(224) | ×(64) | ×(68) | ×(109) |
| 16b78196 | × | × | × | × | × | ×(210) | ×(107) | ×(275) | ×(488) | ×(107) | ×(174) | ×(460) |
| 212895b5 | × | × | × | × | × | ×(317) | ×(153) | ×(623) | ×(1424) | ×(115) | ×(115) | ×(252) |
| 25094a63 | × | × | × | × | × | ×(249) | ×(174) | ×(675) | ×(1344) | ×(62) | ×(171) | ×(460) |
| 256b0a75 | × | × | × | × | × | ×(140) | ×(116) | ×(209) | ×(340) | ×(77) | ×(155) | ×(455) |
| 3ed85e70 | × | × | × | × | × | ×(249) | ×(83) | ×(289) | ×(457) | ×(84) | ×(270) | ×(472) |
| 40f6cd08 | × | × | × | × | × | ×(104) | ×(73) | ×(230) | ×(233) | ×(106) | ×(268) | ×(471) |
| 47996f11 | × | × | × | × | × | ×(321) | ×(147) | ×(794) | ×(1632) | ×(239) | ×(511) | ×(101) |
| 4b6b68e5 | × | × | × | × | × | ×(215) | ×(145) | ×(449) | ×(717) | ×(57) | ×(145) | ×(340) |
| 52fd389e | ✓ | × | × | × | × | ×(209) | ×(94) | ×(373) | ×(633) | ×(89) | ×(202) | ×(368) |
| 79fb03f4 | × | × | × | × | × | ×(280) | ×(102) | ×(1436) | ×(445) | ×(70) | ×(230) | ×(706) |
| 891232d6 | ✓ | × | × | × | × | ×(833) | ×(187) | ×(546) | ×(1468) | ×(84) | ×(276) | ×(257) |
| 896d5239 | × | × | × | × | × | ×(295) | ×(95) | ×(480) | ×(668) | ×(249) | ×(70) | ×(141) |
| 8b28cd80 | × | × | × | × | × | ×(213) | ×(73) | ×(197) | ×(325) | ×(99) | ×(67) | ×(93) |
| 93c31fbe | × | × | × | × | × | ×(149) | ×(141) | ×(527) | ×(741) | ×(76) | ×(70) | ×(141) |
| a3f84088 | ✓ | ✓ | × | × | × | ×(152) | ×(117) | ×(269) | ×(329) | ×(91) | ✓(266) | ✓(759) |
| aa4ec2a5 | ✓ | × | × | × | × | ×(128) | ×(100) | ×(368) | ×(588) | ×(100) | ✓(161) | ×(462) |
| ac0c5833 | × | × | × | × | × | ×(187) | ×(143) | ×(561) | ×(861) | ×(63) | ×(206) | ×(363) |
| b457fec5 | ✓ | × | × | × | × | ×(229) | ×(105) | ×(369) | ×(442) | ×(88) | ✓(145) | ×(343) |
| b7999b51 | ✓ | × | ✓ | × | × | ×(106) | ×(50) | ×(220) | ×(274) | ×(96) | ×(61) | ×(487) |
| b9630600 | × | × | × | × | × | ×(246) | ×(181) | ×(547) | ×(756) | ×(80) | ×(268) | ×(473) |
| c6e1b8da | × | × | × | × | × | ×(151) | ×(71) | ×(363) | ×(305) | ×(83) | ×(112) | ×(247) |
| d931c21c | × | × | × | × | × | ×(176) | ×(81) | ×(326) | ×(438) | ×(71) | ×(264) | ×(735) |
| d94c3b52 | × | × | × | × | × | ×(123) | ×(74) | ×(373) | ×(304) | ×(138) | ×(116) | ×(260) |
| da515329 | × | × | × | × | × | ×(195) | ×(50) | ×(208) | ×(202) | ×(63) | ×(141) | ×(368) |
| e619ca6e | × | × | × | × | × | ×(166) | ×(71) | ×(292) | ×(422) | ×(81) | ×(236) | ×(383) |
| e681b708 | × | × | × | × | × | ×(198) | ×(117) | ×(457) | ×(733) | ×(67) | ×(159) | ×(471) |
| e1d2900e | × | × | × | × | × | ×(189) | ×(44) | ×(521) | ×(622) | ×(83) | ×(197) | ×(556) |
| f3b10344 | ✓ | × | × | × | × | ×(172) | ×(113) | ×(318) | ×(501) | ×(72) | ✓(257) | ✓(671) |
| f9d67f8b | × | × | × | × | × | ×(280) | ×(100) | ×(316) | ×(434) | ×(147) | ×(511) | ×(101) |
| **Avg Time** | × | × | × | × | × | 215 | 109 | 426 | 593 | 99 | 192 | 378 |

Table 2: Ablation experiments on difficult ARC problems that defeat 948 humans.

| Task ID | max | g1.5 | g2.0 | c3.5-ha | c3-ha | c-son | dsv3 | dsr1 | o1-prev | o1mini | o1low | o1med | o1high | o3low | o3high | BARC | MARC |
|---|---|---|---|---|---|---|---|---|---|---|---|---|---|---|---|---|---|
| 31d5ba1a | ✓ | × | × | × | × | × | × | ✓ | ✓ | ✓ | ✓ | ✓ | ✓ | ✓ | ✓ | ✓ | ✓ |
| 79fb03f4 | × | × | × | × | × | × | × | × | × | × | × | × | × | × | × | × | × |
| 8719f442 | ✓ | × | × | × | × | × | × | × | × | × | × | ✓ | ✓ | ✓ | ✓ | × | × |
| a8610ef7 | ✓ | × | × | × | × | × | × | × | × | × | × | × | × | × | ✓ | ✓ | × |
| b4a43f3b | ✓ | × | × | × | × | × | × | × | × | × | × | × | ✓ | ✓ | ✓ | × | × |

# 4   Conclusion

We show that combining diverse inference models and methods with near-perfect verifiers enhances LLMs' performance for advanced reasoning tasks of ARC puzzles. State-of-the-art models, before reasoning models, did not exceed human level in solving tasks with limited training data. However, incorporating LLMs with reasoning by supervised fine-tuning and reinforcement learning enabled models to surpass the average human level performance. By aggregating all models, we increase performance beyond human level. Our approach using diverse models and methods is successful not only in enhancing the success rate compared to individual models, but also solves difficult ARC

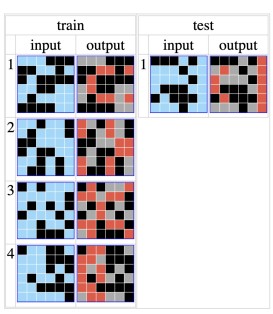

Figure 8: ARC task a8610ef7 on which 948 humans fail and a model or method succeeds.

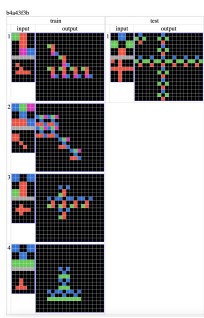

Figure 9: ARC task b4a43f3b on which 948 humans fail and a model or method succeeds.

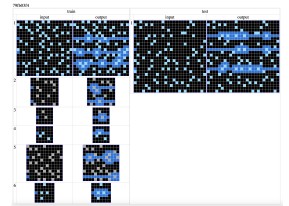

Figure 10: ARC task 79fb03f4 on which 948 humans fail and models or methods fail.

Table 3: ARC model and method performance on evaluation dataset of 400 puzzles.

| Task ID | max | g1.5 | g2.0 | c3.5-ha | c3-ha | c-son | dsv3 | dsr1 | o1-prev | o1mini | o1low | o1med | o1high | o3low | o3high | BARC | MARC |
|---|---|---|---|---|---|---|---|---|---|---|---|---|---|---|---|---|---|
| correct | 373 | 12 | 52 | 34 | 21 | 78 | 53 | 80 | 85 | 52 | 97 | 127 | 155 | 331 | 366 | 212 | 190 |
| % correct | 93.75 | 3 | 13 | 8.5 | 5.25 | 19.5 | 13.25 | 20 | 21.25 | 13 | 24.25 | 31.75 | 38.75 | 82.75 | 91.5 | 53 | 47.5 |

tasks that neither reasoning LLMs nor 948 humans can solve. We demonstrate that using diverse models and methods increases performance in addition to increasing the size of training data and increasing inference time.

## 4.1 Limitations

Our approach assumes access to multiple models (some closed) and incurs additional compute. The simple verifier may occasionally overfit to training pairs, and, as with prior ARC work, our evaluation is limited to the public 400-task test set. Finally, while we solve some tasks humans collectively miss, we do not claim human-level abstraction in general.

## 5 AI Agent Setup

As described in the checklist, this work is mostly human and is assisted by AI. We use OptiLLM [23], which implements diverse methods for LLM inference. We use Claude Code and Co-Pilot as coding assistants and GPT 5 Pro for reviewing. Coding and reviewing are performed iteratively with human oversight and alignment. Next, we plan to use OpenEvolve [30] to discover new algorithms for solving ARC puzzles.

## 6 Ethics and Broader Impact

Solving program-induction puzzles has a positive impact on scientific discovery tools and symbolic reasoning systems. Risks include over-claiming general intelligence from benchmark gains and potential misuse of automated program synthesis. We discuss mitigation in the checklist.

## 7 Reproducibility Statement

We evaluate on the 400 ARC public evaluation tasks using a verifier that executes candidate programs on the provided training pairs, and we count a task as solved if any of 16 diverse methods returns a verified solution. We release drivers and modular implementations (BARC, MARC, MCTS, plan search, mixture-of-agents, self-consistency, best-of-N, etc) and instructions to recreate submission.json files from public ARC JSONs; logs and intermediate artifacts can be placed in the provided folder to replay aggregation. We specify environment details and seeds, and document dependencies. All experiments use only public ARC evaluation data; no private test items are used.

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

## Agents4Science AI Involvement Checklist

This checklist explains the role of AI in the research. The scores for AI involvement are:

- **[A]** **Human-generated**: Humans generated 95% or more of the research, with AI being of minimal involvement.
- **[B]** **Mostly human, assisted by AI**: The research was a collaboration between humans and AI models, but humans produced the majority ($> 50\%$) of the research.
- **[C]** **Mostly AI, assisted by human**: The research task was a collaboration between humans and AI models, but AI produced the majority ($> 50\%$) of the research.
- **[D]** **AI-generated**: AI performed over 95% of the research. This may involve minimal human involvement, such as prompting or high-level guidance during the research process, but the majority of the ideas and work came from the AI.

1. **Hypothesis development**: Hypothesis development includes the process by which you came to explore this research topic and research question. This can involve the background research performed by either researchers or by AI. This can also involve whether the idea was proposed by researchers or by AI.

   Answer: **[B]**

   Explanation: The research idea, that increasing diversity across models/methods improves ARC solve rates, predates this write-up and was proposed and refined by the authors. During this submission, an LLM assistant was used to help tighten the framing, clarifying the claims, and checking for prior related work. The AI did not originate the research question or experimental hypotheses.

2. **Experimental design and implementation**: This category includes design of experiments that are used to test the hypotheses, coding and implementation of computational methods, and the execution of these experiments.

   Answer: **[B]**

   Explanation: All experiments, agent implementations, and verification code were designed and executed by the authors. In this submission process, an LLM was used for minor coding suggestions. Experiments were run by the authors.

3. **Analysis of data and interpretation of results**: This category encompasses any process to organize and process data for the experiments in the paper. It also includes interpretations of the results of the study.

   Answer: **[B]**

   Explanation: Quantitative results were produced by the authors' code and figures where produced by LLMs given these results. The LLM assisted in summarizing and analyzing the results, and checking consistency across the paper.

4. **Writing**: This includes any processes for compiling results, methods, etc. into the final paper form. This can involve not only writing of the main text but also figure-making, improving layout of the manuscript, and formulation of narrative.

   Answer: **[B]**

   Explanation: The LLM contributed to editing: restructuring sections for the Agents4Science format, anonymizing the manuscript, drafting the checklists, polishing language, and generating the LaTeX/ scaffolding. The authors reviewed and revised the text and are responsible for the final content.

5. **Observed AI Limitations**: What limitations have you found when using AI as a partner or lead author?

   Description: The LLM may propose plausible but incorrect citations or mis-state numbers if not grounded in web search. We constrained the paper to author-provided figures, required exact numbers to match logs, and subjected all generated text to human review.

