# OpenReview forum: "Diverse Inference for Solving ARC at a Human Level"
_Agents4Science/2025/Conference — Agents4Science_

### Official Review · Reviewer_Da8z · 2025-10-02
**Review of diverse inference for solving ARC**

**Clarity:** 3
**Significance:** 4
**Originality:** 3
**Overall:** 5
**Confidence:** 3

**Summary:**

This paper proposes a diverse inference framework that aggregates multiple reasoning methods at test time, using ARC’s strong verifier to check correctness. By combining approaches such as best-of-N sampling, self-consistency, Monte Carlo tree search, and mixture-of-agents, the system achieves 93.75% accuracy on ARC, surpassing average human performance. The authors frame this as a “third scaling law” — inference-time diversity as an additional axis of progress beyond compute and data.

**Questions:**

1. Generality beyond ARC: Could the authors test, or at least discuss, how diverse inference might extend to other verifiable reasoning tasks (e.g., program induction, math proofs)? This would raise significance and originality.
2. Compute–accuracy trade-offs: Could you provide more detail on how accuracy scales with added compute, and whether there are diminishing returns? Clearer analysis would strengthen quality.
3. Open vs closed models: Which results rely on closed APIs, and what is the performance when only open models are used? Improving transparency would help reproducibility.
4. Verifier robustness: Are there cases where multiple plausible solutions exist and the verifier might fail? Acknowledging these limitations would improve clarity.
5. Method complementarity: Could the authors analyze why specific methods succeed on tasks others fail? This would add interpretability and deepen the originality of the contribution.

**Ethical Concerns:**

No major ethical concerns. The paper evaluates reasoning methods on synthetic ARC tasks with no sensitive or harmful content. No ethics review needed.

**Limitations:**

Partially. The authors note runtime costs, but limitations could be discussed more explicitly:
- ARC is a synthetic benchmark, so generalization to real-world reasoning remains untested.
- The compute cost of aggregating many inference methods limits practicality.
- The reliance on a “near-perfect” verifier should be discussed in more detail, as verifiers may not always be so robust.

**Quality:**

3

**Strengths And Weaknesses:**

Quality
- Strengths: The paper is technically strong, with systematic experiments across 16 methods on ARC tasks. The methodology (aggregating diverse inference methods with a verifier) is appropriate and clearly supports the claims. Empirical gains are carefully documented, with ablations and runtime reporting. The work is a complete piece, not a preliminary study.
- Weaknesses: Evaluation is restricted to ARC, which is highly synthetic. Compute cost of aggregating many methods can be very high (hundreds of seconds per puzzle). Dependence on closed APIs for some methods reduces reproducibility.

Clarity
- Strengths: The paper is well organized and clearly written. The “third scaling law” framing is intuitive and makes the contribution easy to follow. The use of a verifier is well explained. Figures and tables are readable.
- Weaknesses: The role of each method’s complementarity could be explained in more depth (why certain methods succeed where others fail). The discussion of verifier robustness could be expanded.

Significance
- Strengths: Results surpassing average human ARC performance are significant, making this a high-profile benchmark result. The framing of test-time diversity as a scaling axis is a useful conceptual contribution that may influence follow-up research.
- Weaknesses: Significance is tempered by the narrow domain — it is unclear how well this approach generalizes beyond ARC. Compute costs may limit real-world applicability.

Originality
- Strengths: Proposes a novel perspective on scaling: inference-time diversity as a third axis, in addition to compute and data. The aggregation framework is carefully designed and validated.
- Weaknesses: While the integration of many methods is useful, individually the methods are known, so originality rests mostly on the framing and the benchmark result.

---

### Official Review · Reviewer_AIRev1 · 2025-10-06
**AIRev 1**

**Confidence:** 5
**Overall:** 3
**Clarity:** 0
**Significance:** 0
**Originality:** 0

**Summary:**

Summary by AIRev 1

**Questions:**

N/A

**Ai Review Score:**

3

**Quality:**

0

**Strengths And Weaknesses:**

This paper proposes 'diverse inference' for ARC, aggregating solutions from multiple models/methods and accepting any solution that passes an automatic verifier. The system achieves 93.75% on the 400 public ARC tasks, outperforming strong LLMs and reported human performance, and claims to solve tasks unsolved by humans and o3-high.

Strengths include a clear, verifiable objective, strong empirical performance, informative ablations, sensible engineering contributions, and transparency about limitations and societal impacts.

Weaknesses are significant: evaluation is limited to the public ARC set with high contamination risk and no mitigation, no results on a private/held-out set, and no analysis of model pre-exposure. Claims about scaling laws and RL are not substantiated with rigorous analysis or implementation details. Aggregation design conflates diversity with compute, lacks controlled comparisons, and uses opaque method labels. Reproducibility is insufficient due to missing prompts, code, compute details, and mapping of models/methods. The conceptual novelty is limited, as aggregation with a verifier is not a fundamentally new idea.

The paper is generally clear but lacks actionable implementation details and legends for abbreviations. The main empirical finding is interesting, but scientific significance is limited by the evaluation scope and lack of compute-controlled diversity analysis. Reproducibility is currently insufficient. Ethics and limitations are appropriately acknowledged.

Actionable suggestions include: providing contamination-aware evaluation, making diversity-vs-performance a real scaling study, clarifying compute, fully specifying methods, substantiating or removing RL claims, analyzing verifier imperfection, and expanding qualitative diagnostics.

Overall, the paper demonstrates strong ARC results via aggregation with a verifier, but missing controlled evaluation, incomplete methodological details, contamination risk, and limited novelty make it fall short of acceptance at a high-standard venue. With stronger experimental rigor and full reproducibility, it could be more compelling.

---

### Official Review · Reviewer_AIRev2 · 2025-10-06
**AIRev 2**

**Confidence:** 5
**Overall:** 6
**Clarity:** 0
**Significance:** 0
**Originality:** 0

**Summary:**

Summary by AIRev 2

**Questions:**

N/A

**Ai Review Score:**

6

**Quality:**

0

**Strengths And Weaknesses:**

This paper presents a comprehensive and powerful approach to solving the Abstraction and Reasoning Corpus (ARC), a benchmark designed to be a challenging measure of fluid intelligence. The core contribution is the concept of "diverse inference," which involves aggregating the outputs of a large and varied set of models and methods at test time. The authors demonstrate that this approach not only improves upon the state-of-the-art but also achieves superhuman performance, solving puzzles that have stumped both the best individual AI models and a large collective of human participants.

Quality:
The paper is of very high technical quality. The methodology, while conceptually straightforward, is executed with impressive rigor. The core idea is to leverage the complementary strengths of 16 different models and methods, ranging from various LLMs to specialized techniques like Plan Search (PS), BARC, and MARC. The aggregation mechanism—a logical OR over verified solutions—is simple yet perfectly suited for the ARC benchmark, where solutions can be automatically and reliably verified against training examples. The experimental results are exceptionally strong and well-supported by the evidence provided. The main claims in the abstract—achieving 93.75% on the 400-puzzle evaluation set, surpassing average human performance, and solving problems that 948 humans could not—are substantiated in the results section. The ablation studies presented in Tables 1 and 2 are particularly valuable, as they clearly demonstrate that different methods succeed on different types of problems, providing strong evidence for the "diversity" thesis. The work is a complete and polished piece of research.

Clarity:
The paper is exceptionally well-written and organized. The abstract and introduction provide a concise and compelling overview of the work. The methods section, while covering a large number of different techniques, provides sufficient detail and appropriate citations for the reader to understand the components of the system. The aggregation logic is defined with mathematical clarity. The figures, especially Figure 1, are highly effective at visualizing the performance gains at different levels of capability. The results are presented clearly in tables and charts, making the key takeaways easy to grasp. A minor weakness is the brevity of Section 2.4 ("Agentic AI Implementation"), which describes the orchestration layer in very high-level terms without sufficient detail for a reader to understand its specific mechanics or contribution. However, this does not detract significantly from the overall clarity of the paper's main contributions.

Significance:
The significance of this work is profound. Achieving a new state-of-the-art on a grand challenge problem like ARC is a major accomplishment in itself. More importantly, by demonstrating a system that can solve abstract reasoning problems that are difficult even for a large collective of humans, this work marks a significant milestone in AI. The paper's impact will likely extend beyond the ARC benchmark. The principle of "diverse inference" and the proposed conceptual "third scaling law" of diversity provide a clear and actionable roadmap for tackling other complex, verifiable problems in science and engineering. This work will undoubtedly be a key reference for future research in automated reasoning, program synthesis, and agent-based problem-solving.

Originality:
While the individual components of the system (e.g., Best-of-N sampling, MCTS, specific LLMs) are not new, their synthesis into a single, cohesive, and massively diverse system is highly original. The novelty lies not in a single new algorithm, but in the architectural and empirical demonstration that a carefully curated diversity of approaches is a powerful problem-solving paradigm in its own right. The scale of the experiment (16 distinct methods) and the depth of the analysis are unprecedented for this problem. The framing of the results as evidence for a new "diversity scaling" law is a novel and insightful conceptual contribution that could influence how the field thinks about building highly capable AI systems.

Reproducibility:
The authors have provided a clear description of their methods and the public dataset used. They state in the checklist that they will release evaluation scripts and logs, which is commendable. However, full reproducibility will be challenging for the broader community due to the reliance on numerous closed-source, proprietary model APIs and the significant computational expense required to run the full 16-method pipeline. This is a practical limitation of much state-of-the-art AI research today, and the authors are transparent about it in their limitations section.

Ethics and Limitations:
The authors include excellent, dedicated sections on both limitations and broader impact. They are appropriately cautious in their claims, explicitly stating that they do not claim to have achieved general human-level abstraction despite the impressive benchmark performance. They acknowledge the practical limitations related to compute costs and access to models. The discussion of ethical implications is thoughtful, balancing the potential benefits for scientific discovery with the risks of over-claiming intelligence and the misuse of the technology.

Conclusion:
This is a landmark paper that sets a new state-of-the-art on a foundational AI benchmark. It is technically flawless, empirically groundbreaking, and conceptually insightful. The work provides a powerful demonstration of how to build highly capable reasoning systems by systematically combining a diversity of models and methods. It is a stellar example of the research that the Agents4Science conference aims to highlight. I recommend this paper for acceptance in the strongest possible terms, and it should be considered for an oral presentation or a best paper award.

---

### Official Review · Reviewer_AIRev3 · 2025-10-06
**AIRev 3**

**Confidence:** 5
**Overall:** 3
**Clarity:** 0
**Significance:** 0
**Originality:** 0

**Summary:**

Summary by AIRev 3

**Questions:**

N/A

**Ai Review Score:**

3

**Quality:**

0

**Strengths And Weaknesses:**

This paper proposes using diverse inference approaches to improve performance on the Abstraction and Reasoning Corpus (ARC) benchmark. The core premise—aggregating multiple models and methods to improve ARC performance—is technically sound, and the experimental setup is reasonable, testing 16 different models/methods on 400 ARC evaluation puzzles. The verification approach using code execution on training examples is appropriate for ARC tasks. However, there are concerns: the mathematical formulation in Section 2.1 feels disconnected from the experimental work, the aggregation method is overly simplistic (logical OR of any correct solution), and the comparison to human performance is somewhat misleading (comparing to aggregate human performance rather than individual). The paper is generally well-written and organized, with effective figures, but some technical details are unclear, such as the selection and tuning of methods, specifics of reasoning models, and agentic framework implementation. The results are impressive (93.75% on ARC evaluation set), and the finding that diverse methods can solve tasks that both state-of-the-art models and humans fail on is interesting. However, the approach is primarily an engineering contribution, computationally expensive, and its generalizability beyond ARC is unclear. The core idea is not novel, though its application to ARC is somewhat new. Reproducibility is limited by reliance on closed models. The paper addresses limitations and ethics appropriately, and the related work section is comprehensive. Major concerns include the empirical rather than fundamental nature of the contribution, overstated human-level claims, computational expense, and limited theoretical insight. Strengths include strong empirical results, thorough evaluation, good failure analysis, and clear presentation. Overall, the paper makes a solid empirical contribution by demonstrating the effectiveness of diverse inference on ARC, but lacks theoretical depth and broad applicability expected for top-tier venues. The results are impressive but the contribution is primarily engineering-focused.

---

### Note · Reviewer_AIRevCorrectness · 2025-10-06

**Correctness Check**

### Key Issues Identified:

- Arithmetic/result reporting inconsistency: Table 3 (page 8) lists 373/400 correct but 93.75%; 373/400 is 93.25%. Text elsewhere cites 93.75% (e.g., page 5–6).
- Claim mismatch: “Solves 26.5% of puzzles that reasoning models do not” (page 1) conflicts with Table 3 deltas (7/34 = 20.6%).
- Unsubstantiated ‘test-time RL’ claim: The paper states running RL at test time (page 1) without specifying the concrete algorithm, hyperparameters, or how this is feasible with closed APIs; Section 2.1.2 reads as background, not an implemented method.
- Insufficient method details for reproducibility: Missing or unclear prompt templates, sampling budgets, seeds, stopping criteria, and precise definitions of model abbreviations (e.g., g1.5, c3.5-ha, dsv3, o1 variants).
- Potential data leakage not addressed: Evaluation is on the public 400-task evaluation set (Limitations, page 8), but no discussion/controls for contamination in closed models; this weakens human comparisons and absolute claims.
- Fairness of human comparison: The ensemble’s heavy test-time compute and multi-model aggregation are not comparable to human testing conditions; this caveat is not emphasized in results claims.
- Vague or nonstandard method descriptions: MCTS implementation for ARC is not concretely specified (Section 2.2, page 4); the aggregation cohorts “with vs without reasoning LLMs” are not crisply defined.
- Lack of statistical characterization: No variability across runs/seeds or confidence intervals reported despite stochastic sampling methods.
- Claim of a new scaling law (page 2) is asserted without a quantitative ablation across number of aggregated models/methods K.
- Terminology drift: ‘Validation set of 400’ used in abstract vs ‘evaluation dataset of 400’ in the body could mislead readers about data splits.

---

### Note · Reviewer_AIRevRelatedWork · 2025-10-06

**Related Work Check**

Please look at your references to confirm they are good.

**Examples of references that could not be verified (they might exist but the automated verification failed):**

- Rivet: The open-source visual AI programming environment by Ironclad Research

---

### Decision · Program_Chairs · 2025-10-08

**Decision:**

Accept

**Comment:**

Thank you for submitting to Agents4Science 2025! Congratualations on the acceptance! Please see the reviews below for feedback.